# Hierarchical Decentralized Federated Learning Framework with Adaptive Clustering: Bloom-Filter-Based Companions Choice for Learning Non-IID Data in IoV

**Siyuan Liu** [1], **Zhiqiang Liu** [1,*], **Zhiwei Xu** [2,3], **Wenjing Liu** [4] and **Jie Tian** [5]

1 College of Information Engineering, Inner Mongolia University of Technology, Hohhot 010051, China; siyuan_liu2022@foxmail.com
2 Haihe Laboratory of Information Technology Application Innovation, Tianjin 300350, China; xuzhiwei2001@ict.ac.cn
3 Institute of Computing Technology, Chinese Academy of Sciences, Beijing 100086, China
4 College of Data Science and Application, Inner Mongolia University of Technology, Hohhot 010051, China; liuwenjing2015@bupt.edu.cn
5 Department of Computer Science, New Jersey Institute of Technology, Newark, NJ 07102, USA; jt66@njit.edu
* Correspondence: liuzq@imut.edu.cn

**Abstract:** The accelerating progress of the Internet of Vehicles (IoV) has put forward a higher demand for distributed model training and data sharing in vehicular networks. Traditional centralized approaches are no longer applicable in the face of drivers' concerns about data privacy, while Decentralized Federated Learning (DFL) provides new possibilities to address this issue. However, DFL still faces challenges regarding the non-IID data of passing vehicles. To tackle this challenge, a novel DFL framework, Hierarchical Decentralized Federated Learning (H-DFL), is proposed to achieve qualified distributed training among vehicles by considering data complementarity. We include vehicles, base stations, and data center servers in this framework. Firstly, a novel vehicle-clustering paradigm is designed to group passing vehicles based on the Bloom-filter-based compact representation of data complementarity. In this way, vehicles train their models based on local data, exchange model parameters in each group, and achieve a qualified local model without the interference of imbalanced data. On a higher level, a local model trained by each group is submitted to the data center to obtain a model covering global features. Base stations maintain the local models of different groups and judge whether the local models need to be updated according to the global model. The experimental results based on real-world data demonstrate that H-DFL dose not only reduces communication latency with different participants but also addresses the challenges of non-IID data in vehicles.

**Keywords:** Internet of Vehicles; hierarchical decentralized federated learning; in-network compact representation; vehicle clustering

## 1. Introduction

Advanced Internet of Vehicles (IoV) facilitates vehicle-to-everything (V2X) communications in a smart transportation environment, especially improving the efficiency of data transmission among autonomous vehicles [1,2]. For instance, recent research by Ngo et al. [3] has highlighted the effectiveness of cooperative vehicle communication in improving data transmission efficiency among autonomous vehicles. This complements autonomous vehicles beyond their sensors' sight and makes them cooperate with each other to boost road safety, traffic efficiency, and overall driving experience, thereby shaping the future of urban mobility [4,5]. Considering the fact that IoV is a type of open network, the data exchanged among vehicles are vulnerable without ensuring vehicles behave as per the norms. Therefore, data sharing in IoV becomes scarce [6]. Although

advance learning models for smart transportation have been studied extensively, these models cannot be applied in practice due to lack of data available for model training. Even worse, the vehicles in IoV always seem to involve significant dynamics and driver concerns regarding privacy [7]. Traditional centralized methods for model training and data analysis in vehicular networks face significant challenges in addressing data privacy concerns, as they necessitate the aggregation of sensitive data from multiple vehicles in a single location, exposing individuals to security risks and legal complexities [8]. Moreover, such approaches can compromise data trustworthiness, introduce communication latency, and struggle with scalability. Ultimately, traditional centralized training methods are no longer sufficient to meet the requirements of model training and applications in V2X [9,10]. Those node-centric learning frameworks become ill-fit in IoV.

As a type of federated learning (FL) [11], Decentralized federated learning (DFL) provides a decentralized learning approach for model training while maintaining data at a local level. DFL allows models to be trained on local participants, and they only share model parameters with each other. The shared parameters are also aggregated in each participant [12]. DFL has been widely applied in the industrial Internet of Things (IoT) environment, particularly in the context of vehicular networking [13]. With the advent of autonomous and semi-autonomous vehicles, the need for fast response and real-time decision-making has become more pressing. As emphasized by Soto et al. [14], V2X interactions are instrumental in coordinating vehicles and managing traffic flow, vital components in achieving efficient urban transportation. This dynamic landscape underscores the significance of communication and collaboration among vehicles, especially underlining the importance of DFL. Through DFL, vehicles can conduct model training and inference locally, reducing communication latency with centralized servers and improving responsiveness and real-time capability [15]. However, due to differences in vehicle types and features, the distribution of the training data of passing vehicles may vary greatly. The varying and imbalanced distribution of data among vehicles introduces biases, compromises accurate insights, increases communication overhead, fosters unfair resource allocation, and diminishes real-time performance. This skewed data distribution challenges traditional learning methods and hinders their ability to generalize and adapt effectively [16]. The fluctuating distribution of training data held by passing vehicles hinders traditional learning methods from achieving optimal performance. It is a significant challenge to achieve qualified learning on the non-IID data of vehicles by decentralized federated learning [17].

Addressing the challenges posed by the diverse distribution of vehicle data, a prospective strategy is to leverage the clustering of vehicles based on their complementary data and subsequently implement balanced learning by aggregating model updates within each cluster. While this intra-cluster aggregation ensures learning quality, it may lead to model convergence only with a part of features. To cover all valuable features, a global level of aggregation becomes indispensable. In this way, it is necessary to develop a hierarchical aggregation strategy that leverages vehicle data for distributed model training. In this work, we propose Hierarchical Decentralized Federated Learning (H-DFL), a new framework that incorporates vehicle clustering and hierarchical federated learning through a consideration of data balancing (See Figure 1). The proposed H-DFL approach aims to bridge the gaps left by conventional methods by strategically leveraging vehicle clustering, hierarchical aggregation, and data complementarity. The main contributions are as follows:

1.  To accommodate the evolving intelligent transportation scenarios, we introduce a Hierarchical Decentralized Federated Learning framework, H-DFL, for the distributed training of vehicles in IoV.
2.  We explore a novel paradigm of vehicle clustering according to the data complementarity of vehicles. We take merge-CBF [18] to represent local data in a compact way, in order to support exchange compact representations and judge the data complementarity of other vehicles. Finally, we cluster vehicles with complementary data into one group and train local models with the group.

3.  On a higher level, base stations maintain the local models of different groups, while submitting them to a data center to obtain a global aggregation to cover more features. In an asynchronous manner, base stations judge whether the local models need to be updated according to the global model.
4.  We evaluate our proposed method and baselines using two real-world datasets on the Sim4DistrDL simulation platform. The simulation results validate the effectiveness and superiority of our approach against existing baselines.

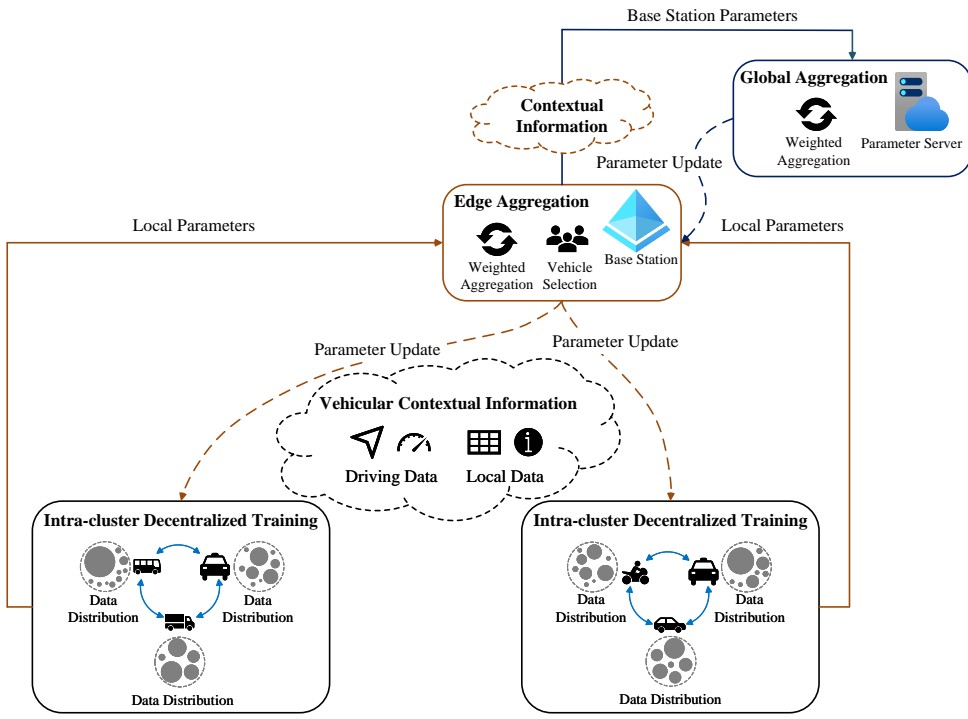

**Figure 1.** Hierarchical decentralized federated learning.

The organizational structure of this paper is as follows. Section 2 reviews the relevant literature on participant selection strategies in federated learning. Section 3 provides a detailed introduction to the relevant theoretical methods. In Section 4, a Hierarchical Decentralized Federated Learning framework is proposed, which adopts a novel paradigm of local clustering to mitigate the impact of imbalanced data. In Section 5, two real-world datasets are utilized to demonstrate the superiority of the proposed framework in the V2X scenario compared to the current state-of-the-art techniques. Finally, a summary of this work is presented in Section 6.

## 2. Related Work

In recent years, Federated Learning (FL) has emerged as a distributed optimization paradigm that enables collaboration in model training among a large number of client nodes with limited resources, without sharing data. However, FL faces several challenges, such as heterogeneity among clients, label noise, and biased client selection. To address these challenges, researchers have conducted a series of explorations.

In reference [19], Lim et al. presents a framework that addresses communication inefficiency challenges in the FL network. This approach leverages a Hierarchical Federated Learning (HFL) framework with cluster heads to facilitate data aggregation and model training on heterogeneous distributed devices. In reference [20], Lim et al. focus on addressing node failures and device dropouts through a Hierarchical Federated Learning (HFL) framework, introducing a two-level resource allocation and incentive mechanism involving cluster selection dynamics modeled by evolutionary game theory and a deep-learning-based auction mechanism, ultimately demonstrating the effectiveness of these

mechanisms in terms of uniqueness, stability, and revenue optimization. Considering the problem of malicious node behaviors, Li et al. [21] present a novel DFL framework, Trustiness-based Hierarchical Decentralized Federated Learning (TH-DFL), which incorporates a trust mechanism through Security Robust Aggregation (SRA) to enhance privacy protection and robustness against malicious nodes, while also reducing communication overhead.

In reference [22], Cho et al. introduced a client selection method based on discriminative weights. This strategy reduces the number of transmitted models by selecting clients with significant weight updates, thereby reducing communication overhead. In reference [23], Ribero and Vikalo proposed an optimal sampling strategy based on weight updates. This strategy models the progress of model weights and selects clients with significant weight updates, reducing the number of transmitted models and improving communication efficiency. Cho et al. presented an adaptive client selection strategy in reference [24] to adapt to FL on mobile participants. This strategy selects participating clients based on activity status and mobility, achieving smooth model updates when participants join or leave. Ruan et al. extended the traditional FL paradigm to allow flexible participant participation in reference [25]. They proposed a novel joint aggregation scheme that can adapt to different participant states, achieving model convergence even when participants may be inactive or provide incomplete updates. Ji et al. explored the client selection problem in privacy-preserving settings in [26]. They proposed a random selection strategy that takes privacy requirements into account, thus protecting client data privacy. Liu et al. presented FedEraser [27], which efficiently removes specific training data from a trained FL model. They reconstruct the unlearning model by leveraging historical parameter updates preserved on the central server during the FL training process.

In actual edge application scenarios, the data of edge participants is heterogeneous. Several studies focus on formulating client selection strategies based on client data. In reference [16], Nishio et al. proposed a client selection method based on diversity and representativeness. This strategy involves selecting clients with diverse and representative data to participate in model updates, aiming to achieve better global model performance. In reference [28], Goetz et al. considered scenarios where clients have different data distributions. By designing appropriate client selection strategies and combining methods for handling non-iid data, they achieved model updates and convergence in such non-iid environments. Cho et al. investigated biased client selection in FL [29] and found that biasing client selection towards clients with higher local losses can accelerate error convergence. Based on this observation, they proposed Power-of-Choice, a communication- and computation-efficient client selection framework that offers flexible trade-offs between convergence speed and solution bias. Tan et al. proposed a novel FL framework called FedProto in reference [30], which enhances tolerance to heterogeneity by aggregating knowledge among clients through a communication abstract prototype instead of gradients. Fang and Ye [31] addressed robust FL problems with noise and heterogeneous clients. They proposed the RHFL framework, which effectively mitigates the negative impact under different noise levels/types by directly aligning model feedback using public data, applying robust noise-tolerant loss functions, and designing client confidence reweighting schemes.

These research works have made valuable explorations in addressing heterogeneity, unlearning, bias, and noise-related issues in FL. They have demonstrated the effectiveness of corresponding methods on various datasets and tasks. However, further research is still needed to continue promoting the development of FL technology and addressing various challenges in practical V2X applications.

## 3. Preliminary

### 3.1. Federated Averaging

Federated averaging is a widely adopted parameter aggregation technique that entails averaging the local model parameters of individual participants to derive the global

model parameters. This approach offers several benefits, including efficiency, scalability, decentralization, and applicability to large-scale distributed systems and privacy-sensitive scenarios. This section presents an overview of the fundamental principles underlying federated averaging.

The Federated Average Algorithm (FedAvg) [11] is commonly utilized in both centralized and hierarchical federated learning. The algorithm assumes the existence of *K* clients and a central server, with the former conducting local training and the latter responsible for parameter aggregation. FedAvg comprises the following steps:

Step 1: Initialization phase. The central server initializes the model and distributes the model parameters to all participating clients.

Step 2: Local update phase. Each client employs the received model parameters as the starting point for the current iteration and trains on their local dataset using optimization algorithms, such as gradient descent, with the aim of minimizing the loss function. After several iterations, if the client is selected by the central server, the model parameters (e.g., parameters, gradients, etc.) are uploaded to the central server.

Step 3: Global aggregation phase. The central server aggregates the model parameters uploaded by selected clients using a weighted average method to obtain the global model:

$$w_{t+1} \leftarrow \sum_{k=1}^{K} \frac{n_k}{n} w_{t+1}^k, \tag{1}$$

as depicted in Equation (1), where $n$ refers to the size of the dataset owned by the selected client, $n_k$ denotes the size of the local dataset of the $k$-th client, and $w_{t+1}^k$ refers to the parameters of the model of the $k$-th client after $t + 1$ rounds of training. Following parameter aggregation by the central server, the updated model parameters are disseminated to the clients. The client then proceeds to a new round of training according to Step 2. This federated learning process is repeated iteratively until the model converges or reaches the maximum number of training rounds.

*3.2. Consensus Learning*

Consensus learning is a decentralized collaborative algorithm that addresses the parameter aggregation challenge in federated learning by enabling participants to reach a consensus through information exchange without the need for a central server. Consensus learning can enhance the efficiency and security of federated learning, and mitigate potential threats such as malicious attacks and data breaches. In a federated learning system with *N* distributed participants, each node updates its local learning based on local data and information exchanged with neighboring nodes, and repeats the process until all nodes converge to a consensus estimate. In the average consensus algorithm, the *i*-th node updates its parameters in the $k + 1$-th round based on its local training results and those of its neighbors, as follows:

$$\hat{\boldsymbol{\theta}}_i(k+1) = \hat{\boldsymbol{\theta}}_i(k) + \varepsilon \sum_{j \in \mathcal{N}_i} \left( \hat{\boldsymbol{\theta}}_j(k) - \hat{\boldsymbol{\theta}}_i(k) \right) \tag{2}$$

Here, $i$ denotes the node to be updated locally, $k$ denotes the result of the previous round of iterations, $\mathcal{N}_i$ denotes the set of nodes $i$ and its neighbors, $\varepsilon$ denotes the step size, $\hat{\boldsymbol{\theta}}_j(k)$ denotes the model parameters of the neighbor node $j$ at the end of the $k$-th round of learning, and $\hat{\boldsymbol{\theta}}_i(k)$ denotes the model parameters of node $i$ at the end of the $k$-th round of learning.

To approach the global performance of federated learning, the weighted average consensus method introduces a weight in the update rule of the average consensus method. The weighted average consensus update rule is given by:

$$\hat{\boldsymbol{\theta}}_i(k+1) = \hat{\boldsymbol{\theta}}_i(k) + \varepsilon \mathbf{W}_i^{-1} \sum_{j \in \mathcal{N}_i} \left( \hat{\boldsymbol{\theta}}_j(k) - \hat{\boldsymbol{\theta}}_i(k) \right), \tag{3}$$

where $\mathbf{W}_i^{-1}$ represents the weighting coefficient.

### 3.3. Mergeable Counting Bloom Filter

The Bloom Filter [32] is a widely adopted compact data representation structure that provides an efficient solution for data aggregation by saving space. The Counting Bloom Filter (CBF) [33] is a variation of the Bloom Filter that not only determines the presence of elements in a set but also keeps track of the frequency of each element's occurrence. CBF achieves this functionality by leveraging bit array counters instead of binary values alone. Nonetheless, directly merging CBFs is not possible due to the potential risk of overflow issues when combining the bit array counters. To overcome this limitation and aggregate data compact representation of each participant, we use a mergeable CBF called mergeCBF [18] here.

The design overview of mergeCBF is illustrated in Figure 2. In mergeCBF, the counter array of CBF is expanded into a set of bit arrays $barr_1, \ldots, barr_g$, where each bit array has $M$ bits. Following the principles of cuckoo hashing, these bit arrays are scheduled in a fixed order to record the inserted items. When obtaining the hash result for an item, mergeCBF selects one bit array and sets the corresponding cell to "1". This process is repeated $h$ times using different hash functions $Hash_1, \ldots, Hash_h$. After setting $h$ cells based on the $h$ hash results, all the "1" bits in the different bit arrays of mergeCBF's filters are utilized to record the inserted item's information. In this way, by querying these bit arrays, we can determine whether an item exists in mergeCBF and estimate its frequency of occurrence. When multiple mergeCBF instances need to be merged, an efficient way to support the merge is by performing bitwise OR operations on the bit arrays. For the merged result, an additional array is included to record the bitwise OR result of all cells in each column, which is used to query the existence of items. This design facilitates the evaluation of the data distribution of each participant, and groups the participants with complementary data into a group. In this way, the features learned by the group can cover the domain knowledge more comprehensively.

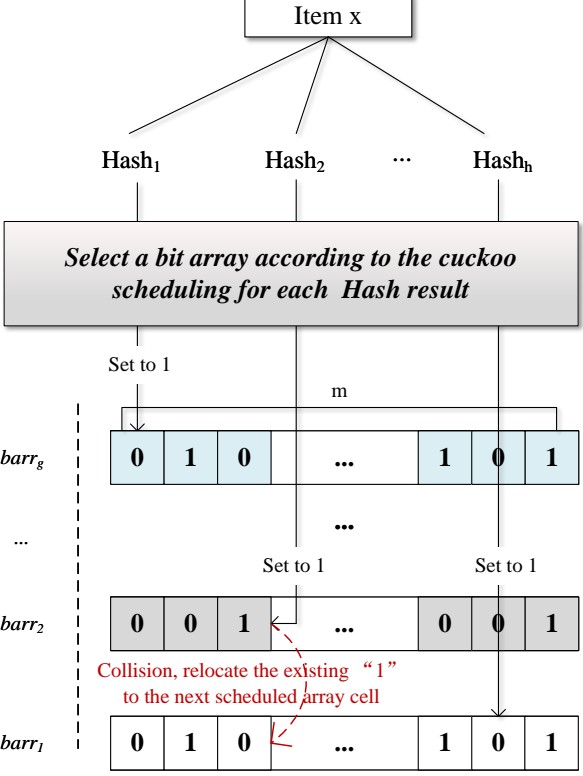

**Figure 2.** Overview of mergeCBF.

## 4. Adaptive Hierarchical Decentralized Federated Learning

### 4.1. Framework

In this subsection, we propose H-DFL, a Hierarchical Decentralized Federated Learning framework with a novel participants clustering approach to mitigate the impact of imbalanced data among participants in the global model. Specifically, we first explore a new scheme for clustering participants, mapping the data into a compact representation to serve as a record of local data distribution, and then clustering participants based on this compact representation to achieve a more uniform distribution of data within each cluster. In Section 4.3, we further present the Hierarchical Decentralized Federated Learning framework, H-DFL, for the distributed training of vehicles within the network.

The algorithm maps the data into a compact representation as a record of local data distribution. By exchanging this type of compact representation, participants are clustered into several clusters with evenly distributed data, thereby reducing the bias in the global model caused by uneven data distribution. After several rounds of decentralized federated learning in each cluster, a representative is randomly selected to perform centralized federated aggregation with the edge server. At a higher level, edge servers perform the federated averaging of parameters with parameter servers to correct for bias. The H-DFL algorithm consists of two main components: participant clustering and global model aggregation. Subsequently, in Sections 4.2 and 4.3, we provide detailed explanations of participant clustering and global model training, respectively.

### 4.2. Vehicle-Side Clustering and Cooperation Based on mergeCBF Filters

In practice, a vast amount of data will be generated in V2X networks. The data collected by vehicles are often influenced by various factors, such as vehicle type, driving conditions, and geographical location. Consequently, the data collected by different vehicles exhibit certain characteristics and variances. These characteristics and variances may lead to biases in the distribution of data labels within the vehicle's dataset, thus affecting the efficiency and accuracy of model training. To address this issue, we propose a Data Distribution Statistics Algorithm (DDSA), a method for clustering vehicles to achieve a uniform distribution of data in a cluster.

MergeCBF is a compact and mergeable data structure that can be used to represent the data distribution of vehicles. Specifically, the parameters such as the size of the bit arrays and the number of hash functions used in mergeCBF are initialized. To represent the sparsity and density of the data distribution, we employ locally sensitive hash functions in mergeCBF. Next, each data from vehicles $k$ is inserted into $mergeCBF_k$. For each data, multiple hash calculations are performed using the specified locally sensitive hash functions, and the corresponding cells in the bit arrays are set to "1" according to the cuckoo scheduling. Finally, the bit array $mergeCBF_k$ recording the data distribution of the vehicles is transmitted to nearby vehicles.

After receiving the bit array of data distribution from neighboring vehicles, vehicles with complementary data will be clustered together. We measure the level of data complementarity among them by merging the mergeCBFs of vehicles. Specifically, to represent the data complementarity, as shown Figure 3, we perform bitwise OR operations on the cells of each column when merging the mergeCBFs. As we employ locally sensitive hash functions, similar data will have similar positions in the array. Therefore, after the bitwise OR operation, the count of "1" in each column can indicate the level of complementarity for the corresponding data. We use an additional array, named $sumBar$, to record the count of "1" in each column.

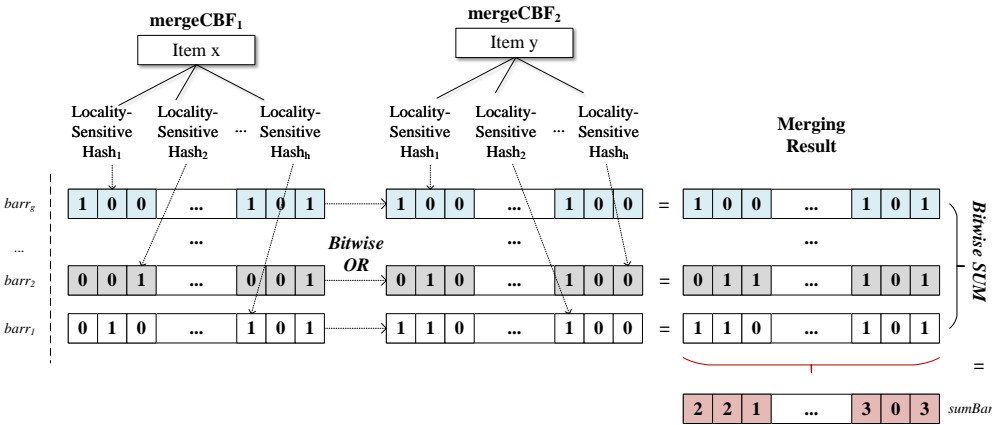

**Figure 3.** Merging CBF.

Clustering is performed to achieve a more uniform distribution of data within groups, and it is essential for the bits of *sumBar* to be closely similar in order to guarantee such uniform distribution. Therefore, we introduce the concept of variance to measure the similarity of each bit in the merged bit array *sumBar*. The formula for calculating the variance *Var* is as follows:

$$Var = \frac{\sum_{m=1}^{M} \left( sumbarr[m] - \overline{sumbarr} \right)^2}{M}, \tag{4}$$

where $m(1, \ldots, M)$ represents the number of cells in *sumbarr*. Ultimately, clustering is carried out based on the magnitude of the variance, and the clustering results are sent back to other neighboring vehicles. Specifically, initially, each vehicle is treated as an individual cluster. By calculating the variance of the merged *sumBar* between each pair of clusters, the complementarity of the distribution between two clusters is determined. The two clusters with the minimum variance are selected, and they are merged to form a new cluster. This process of merging the most similar clusters is repeated until all base stations are clustered. The specific DDSA is shown in Algorithm 1.

---

**Algorithm 1** Data Distribution Statistics

---

1:  **Input:** item, number of clusters $n_{cluster}$;
2:  **Output:** Base station set $Cluster_l$;
3:  $barrset_k = mergeCBF.insert(item_k)$    ▷Insert each data item of vehicle $k$ into mergeCBF.
4:  receive $barrset_{p\{p \in N_k\}}$
5:  **for** each $barrset$ in $barrset_{c\{c \in N_k\}}$ **do**
6:      create $Cluster_l$
7:  **end for**
8:  **while** the number of $Cluster_c \geq n_{cluster}$ **do**
9:      **for** each $Cluster_i$, $Cluster_j$ in $Cluster_c$ **do**
10:         $sumbarr_{ij} = mergeCBF.merge(Cluster_i, Cluster_j)$    ▷Merge the mergeCBF arrays.
11:         $Var_{ij} = Variance(sumbarr_{ij})$    ▷Calculate the variance of *sumbarr*.
12:     **end for**
13:     Find the minist $Var_{ij}$
14:     merge $Cluster_i$ and $Cluster_j$
15: **end while**

---

Algorithm 1 takes the training data of vehicle $k$ and the number of clusters $n_{cluster}$ as inputs, and outputs the clustering of vehicles $Cluster_{c\{c \in N_k\}}$, where $N_k$ represents all

neighboring vehicles. Firstly, mergeCBF is used to count the data distribution of vehicle $k$, resulting in $barrset_k$ (line 3). Then, $barrset_{p\{p\in N_{\bar{k}}\}}$, where $N_{\bar{k}}$ represents the neighboring vehicles except for $k$, is received. Next, each neighboring vehicle is initialized as an initial cluster $Cluster_l$ (lines 5–7). To ensure an even distribution of data in each cluster, the variance of the merged $sumarr_{ij}$ for each cluster is calculated using Equation (4) (lines 9–12). Finally, the clusters are clustered based on the magnitude of variance, merging the two clusters with the smallest variance until the desired number of clusters is achieved. The resulting grouping is then sent to other vehicles.

### 4.3. Global Aggregation Based on Gradient Difference

In this subsection, we introduce and propose a Hierarchical Decentralized Federated Learning (H-DFL) algorithm. Firstly, neighboring vehicles are clustered based on the data distribution of each vehicle, aiming to achieve an even distribution of data within each cluster. Then, decentralized federated learning is applied to form weak learners within each cluster. Each base station randomly selects a vehicle from each neighboring cluster and averages the learning results of vehicles in different clusters to form a strong learner. At the base station, we evaluate the unevenness of data within each cluster and perform weighted averaging based on the evaluation results and send them to both the vehicles and the parameter server. The parameter server performs a weighted averaging of the received parameters and sends the averaged result back to the base station. During the parameter averaging process, network instability may result in delayed parameter transmission for some base stations. Therefore, we adopt an incremental federated averaging aggregation strategy. Due to the typically long transmission distance to the parameter server, the base station does not wait for the averaged result before proceeding to the next training round. Upon receiving the global parameters from the parameter server, the base station compares them with the current parameters. If there is a significant difference, the base station replaces its current parameters with the received global parameters.

The proposed communication framework follows a three-layer communication flow, as follows:

(1) The parameter transmission between base stations and neighboring vehicles

After clustering, decentralized federated learning is conducted within each cluster for a certain number of rounds. The base station randomly selects a vehicle from each neighboring cluster, and the vehicles upload their trained model parameters and the data distribution arrays of all vehicles within the cluster to the base station. We propose an evaluation metric $D_l$ to assess the unevenness of data distribution within each cluster, which is calculated as follows:

$$D_{l\{l\in N_G\}} = 1 \setminus \left( \sum_{m=1}^{M} s_l[m] \cdot \log_2 \frac{s_l[m]}{1\setminus m} \right), s_l[m] = \frac{sumarr_l[m]}{\sum_{m=1}^{M} sumarr_l[m]}. \tag{5}$$

Here, $N_G$ represents the set of vehicles randomly selected by the base station from each cluster, $sumarr_l$ denotes the data distribution of base station $l$ within its respective cluster, and $m(1,\ldots,M)$ represents the length of the $sumarr_l$ array. We can measure whether data distribution within each cluster is balanced. Let $D$ indicate a more balanced data distribution within the cluster. Based on the value of $D$, we perform a weighted average at the base station as follows:

$$w_{t+1} \leftarrow \sum_{l=1}^{L} \frac{n_l}{n} \cdot \frac{D_l}{\sum_{l=1}^{L} D_l} w_{t+1}^l. \tag{6}$$

where $L$ represents the number of vehicles participating in global aggregation, $n_l$ denotes the size of the dataset owned by the $l$-th vehicle in its respective cluster, $n$ represents the size of the cluster dataset participating in global aggregation, and $w_{t+1}^l$ represents the parameters of the model after $t+1$ rounds of training for vehicle $l$.

(2) The parameter transmission between the parameter server and base stations.

After each round of the locally weighted aggregation of nearby vehicle parameters, the base station not only sends the result back to the vehicle cluster, but also sends the result to the parameter server. Since the parameter server is generally far away and the transmission is delayed, the base station does not wait for the global aggregation result returned by the parameter server. An incremental parameter aggregation strategy is employed on the parameter server, and the aggregated parameter results are transmitted back to all base stations. After receiving the global aggregation result $w_{e+1}^{G}$ sent by the parameter server, the base station compares it with the current local weighted aggregation result:

$$w_{e+1}^{G} - w_{t+1}, \tag{7}$$

and replaces the current parameter with the global parameter if the difference is larger than threshold $\beta$.

(3) The parameter transmission between vehicles.

There are two types of parameters transmitted between vehicles: the distribution of vehicles data and the model training parameters. The first parameter is used for clustering vehicles to address the issue of class imbalance. Firstly, Algorithm 1 is employed to compute the data distribution for each vehicle. Then, the computed data distribution results are broadcasted to nearby vehicles in the form of bitmaps. Upon receiving the bit arrays, the vehicles merge the bit arrays and form clusters based on the variance of the merged bit array.

Regarding the second parameter, after each vehicle in a cluster undergoes a certain level of local learning, the parameters are transferred to neighboring nodes. The neighboring nodes converge the parameters based on their local training results and the training results of their neighbors. This process repeats until a consensus is reached within the cluster.

The detailed design of the algorithm and a description of the model parameters are shown in Algorithm 2.

In Algorithm 2, the base station utilizes a random sampling algorithm to select one vehicle from each cluster for centralized federated learning. The set of selected vehicles is denoted as $N_G$, and the participating vehicles in training are represented by $l$. The aggregated parameters from the base station are sent to the parameter server for global aggregation according to Equation (1), where $N_b$ denotes the set of base stations. Upon receiving the globally aggregated parameters from the parameter server, the base station compares them with its current parameters using a threshold value, denoted as $\beta$. If the difference exceeds the threshold value, the base station updates its parameters to the received global parameters. Within each cluster, vehicles engage in decentralized federated learning based on weighted average consensus, where the learning rate is denoted as $\eta_t$ and the weights are represented by $\gamma_t$. Each vehicle locally trains its model using mini-batch gradient descent, with the size of the local mini-batch for participating vehicles denoted as $B$.

The algorithm proposed in this paper is based on vehicle clustering, where vehicles are clustered according to their data distribution. Due to the influence of geographical locations, the overall uniformity of data distribution differs among vehicle clusters. Therefore, we evaluate the uniformity of data within each cluster at the base station and perform a weighted average, as shown in Equation (6). Additionally, the distance between base stations and the parameter server negatively affects the effectiveness of global aggregation. The time interval for each iteration at the parameter server is influenced by the base station with the longest transmission time, leading to a bottleneck effect. Thus, during the global aggregation process, we adopt an incremental parameter update method. Firstly, a global aggregation based on averaging is performed using the received parameters. Once the parameter server completes the aggregation, the result is sent back to the base stations. If there are new base stations sending data at that time, they will be added to the aggregation process and the parameters will be updated according to Equation (6) (lines 8–18). For intra-cluster aggregation, a decentralized method, as described in Equation (3) (lines 19–28), is

employed. Prior to consensus learning within the cluster, multiple local stochastic gradient descents are executed locally to ensure a certain level of accuracy in each base station's local model (lines 29–35).

---

**Algorithm 2** Hierarchical Decentralized Federated Learning Algorithm

---

 1: Parameter Server side:
 2: **for** each global round $e = 1, 2, \ldots, E$ **do**
 3:      **for** each base station $b \in N_B$ **do**
 4:          $w_{e+1}^b \leftarrow GroupUpdate(b, w)$
 5:      **end for**
 6:      $w_{e+1}^G \leftarrow \sum_{b=1}^B \frac{n_b}{n} w_{e+1}^b$
 7: **end for**
 8: Base Station side:
 9: **for** each round $t = 1, 2, \ldots, T$ **do**
10:      $N_G \leftarrow$ (Each group randomly selects one)
11:      **for** each vehicle $l \in N_G$ **do**
12:          $w_{t+1}^l, sumbarr_{t+1}^l \leftarrow GroupUpdate(l, w, sumbarr)$
13:      **end for**
14:      $w_{t+1} \leftarrow \sum_{l=1}^L \frac{n_l}{n} \cdot \frac{D_l}{\sum_{l=1}^L D_l} w_{t+1}^l$     $\triangleright$ Perform global weighted average aggregation.
15:      **if** $w_{e+1}^G - w_{t+1} \geq \beta$ **then**
16:          $w_{t+1} \leftarrow w_{e+1}^G$    $\triangleright$ Update the parameters of base station.
17:      **end if**
18: **end for**
19: Intra-cluster aggregation:
20: **for** each vehicle $k \in N_k$ in parallel **do**
21:      receive $w_{t,i}\{i \in N_{\bar{k}}\}$
22:      $\varphi_{t,k} \leftarrow w_{t,k}$    $\triangleright$The local parameters are updated by receiving the aggregated parameters of the base station.
23:      **for** all vehicles $i \in N_{\bar{k}}$ **do**
24:          $\varphi_{t,k} \leftarrow \varphi_{t,k} + \gamma_t \eta_{t,i}(w_{t,i} - w_{t,k})$    $\triangleright$Update parameters by the weighted average consensus method.
25:      **end for**
26:      $w_{t+1,k} = ModelUpdate(\varphi_{t,k})$
27:      Send $(w_{t+1,k})$
28: **end for**
29: Vehicle gradient update:
30: $VehicleUpdate(\varphi_{t,k})$
31: B $mini-batches$ of size $B$
32: **for** batch $b \in B$ **do**
33:      $\phi_{t,k} \leftarrow \varphi_{t,k} - \eta_t \nabla L_{t,k}(\varphi_{t,k})$
34:      $w_{t,k} \leftarrow \phi_{t,k}$
35: **end for**

---

Informed by the research of El-Ghamrawy [34], we draw upon a novel clustering strategy grounded in Bloom filter techniques to effectively address the challenge of data imbalance within clusters. Inspired by this approach, our proposal capitalizes on the inherent capabilities of Bloom filters to achieve data balance, contributing to more equitable data representations within each cluster. Moving beyond this local level, we extend the concepts of common federated averaging strategies, thereby enabling the amalgamation of high-level parameters to attain a comprehensive global model. By merging the strengths of Bloom-filter-based clustering with established federated learning principles, our proposal seeks to offer a robust and balanced framework for hierarchical decentralized federated learning.

## 5. Experiments

We conducted experiments on the Sim4DistrDL [35], with a three-layer network topology, in order to validate the convergence speed and accuracy of our proposed H-DFL method.

### 5.1. Experimental Details

In this subsection, we present the experimental details of IoV, encompassing comprehensive information regarding the simulation setup and training tasks.

### 5.1.1. Simulation Setup

In order to construct a environment for IoV that closely resembles a real-world scenario, we conducted simulations using Sim4DistrDL [35]. The simulation environment consists of a parameter server, four base stations, and four neighboring vehicles for each base station. Wireless communication base stations serve as roadside participants to facilitate communication within the IoV network. Neighboring vehicles collaborate by using local data to train deep neural network (DNN) models and communicate with the base stations. Simultaneously, the four base stations send parameters to the parameter server and receive updated global parameters in return. Each base station is connected to four vehicles via V2X communication, and the clusters of vehicles change over time in response to fluctuations in local data. Figure 4 illustrates the initial clustering scenario.

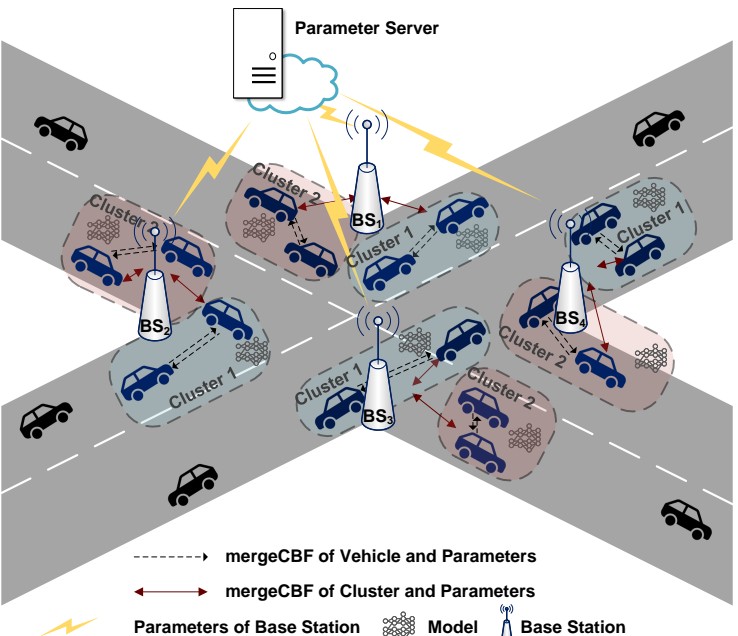

**Figure 4.** The topology structure of the H-DFL vehicle scenario.

### 5.1.2. Training Tasks and Data Redistribution

In order to validate the efficiency of H-DFL, we considered two types of machine learning tasks related to image classification, VGG-16 [36] and Multi-Layer Perceptron(MLP) [37]. The image classification tasks were based on two different datasets, namely the Belgian Traffic Sign Classification Benchmark (BelgiumTSC) [38] and CIFAR-10 [39].

BelgiumTSC consists of more than 7000 high-resolution color images captured from different angles and under various environmental conditions. It provides precise annotation in the form of bounding boxes for each traffic sign present in the images. These bounding boxes indicate the exact location and extent of the traffic signs. In total, there are 11,219 annotated bounding boxes, which correspond to more than 2000 individual traffic sign instances. CIFAR-10 is a popular image classification dataset widely used in the field

of computer vision and machine learning. It consists of a collection of 60,000 small color images and is divided into 10 different classes.

Given that federated learning must account for the heterogeneity in data distributions among participants, it was necessary to create imbalanced datasets among the participants. For this purpose, we selected five classes from the BelgiumTSC, and Figure 5a illustrates the initial data distribution for the four vehicles at each base station during training. Similarly, when using the CIFAR-10 dataset, Figure 5b displays the initial data distribution for the four vehicles at each base station.

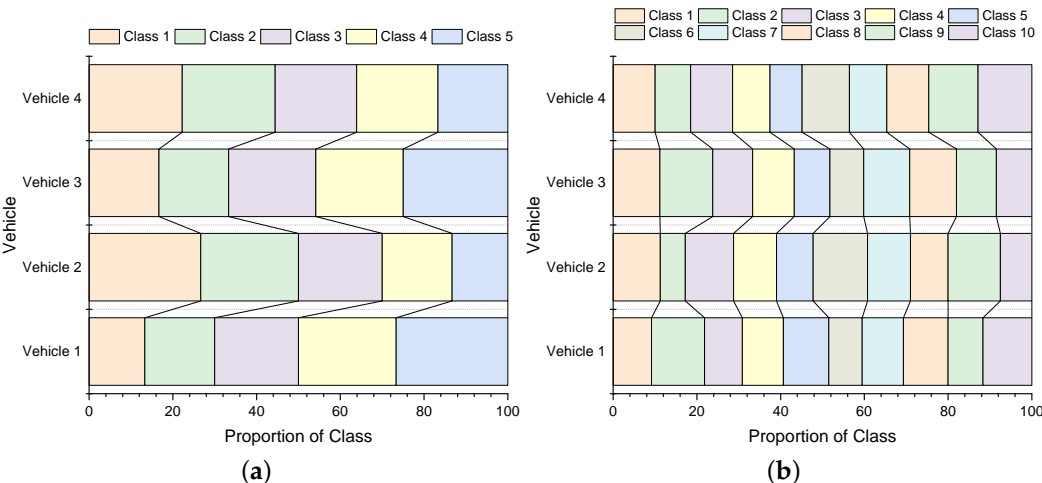

**Figure 5.** The initial data distribution of vehicles. (**a**) The initial data distribution of BelgiumTSC. (**b**) The initial data distribution of CIFAR-10.

### 5.1.3. Baselines

To evaluate the convergence speed and accuracy of H-DFL, we compared three classical algorithms, comparing H-DFL with other three FL protocols: FedAvg [11], Consensus-based FA (CFA) [12], and Cooperative Decentralized Federated Averaging (CDFA) [40].

FedAvg is a distributed machine learning algorithm with a network topology consisting of two layers: participants and parameter servers. After completing local learning on each participant, the parameter server randomly selects a subset of participants to upload their learned model parameters to the server. The parameter server aggregates the model parameters received and sends back the aggregated result to the participants. This process iterates until the model converges.

CFA is a consensus-based decentralized federated learning strategy. It leverages decentralized networks, enabling devices to perform collaborative training steps using local data and consensus-based methods, ensuring convergence efficiency across varying connectivity graphs and network sizes, particularly within industrial IoT settings.

CDFA addresses privacy concerns in smart healthcare, employing real-time distributed networking via MQTT protocol and validating its efficacy through brain tumor segmentation on physically separated machines with varying operational conditions.

### 5.1.4. Evaluation Metrics

To evaluate the performance of the proposed algorithm in this paper, two metrics are used: loss and accuracy. The loss of model training on the dataset is defined as follows:

$$H(y, q) = -\frac{1}{N} \sum_{i=1}^{n} y_i^{(c)} \log\left(q_i^{(c)}\right), \tag{8}$$

where $N$ is the number of iterations required to complete a round of training, $n$ is the total number of categories in the classification, $y_i^{(c)}$ is the value of whether the $i$-th item belongs to class $c$, and $q_i^{(c)}$ is the predicted probability that the $i$-th item belongs to class $c$.

The test accuracy (Acc) of the model on the data set is defined as follows:

$$Acc = \frac{1}{N} \frac{\sum_{i=1}^{n} T_i}{\sum_{i=1}^{n} (T_i + F_i)},$$

(9)

where $n$ is the total number of classes, $T_i$ is the number of correctly classified items in the $i$-th class, and $F_i$ is the number of misclassified items in the $i$-th class.

### 5.2. Performance in Different Training Tasks

In this subsection, we primarily analyze the results of H-DFL and two other Federated Learning algorithms on the two training tasks, focusing on two aspects: convergence speed and accuracy.

For the FedAvg, communication rounds refer to the rounds of communication between participants and the parameter server. Due to the typically large communication distances, significant communication overhead is incurred. In the case of the CFA, the number of communication rounds is calculated based on the communication rounds with neighboring nodes. As for H-DFL, communication rounds encompass both rounds of communication with neighbors and rounds of communication with the base stations. In the experiments, for each vehicle, batch size, learning rate, and the number of epochs for local training were set to 10, 0.01, and 1, respectively.

The convergence speed of the VGG model under different distributed learning algorithms based on the BelgiumTSC dataset is presented in Figure 6. The results indicate that the H-DFL algorithm (black curve) exhibits the fastest convergence, typically starting to converge after around 10 communication rounds. The second fastest convergence is observed in the CFA algorithm (red curve), while the FedAvg (green curve) and CDFA (blue curve) algorithms demonstrate slower convergence rates. Figure 7 displays the loss values' variation of the MLP model on the non-uniform CIFAR-10 dataset. The results suggest that the H-DFL algorithm achieves the fastest convergence, followed by the CFA algorithm and CDFA algorithm, while the FedAvg algorithm exhibits a slower convergence rate. Real-time responsiveness is a critical factor for vehicular networks or IoV applications, and H-DFL appears to be better suited for the IoV environment.

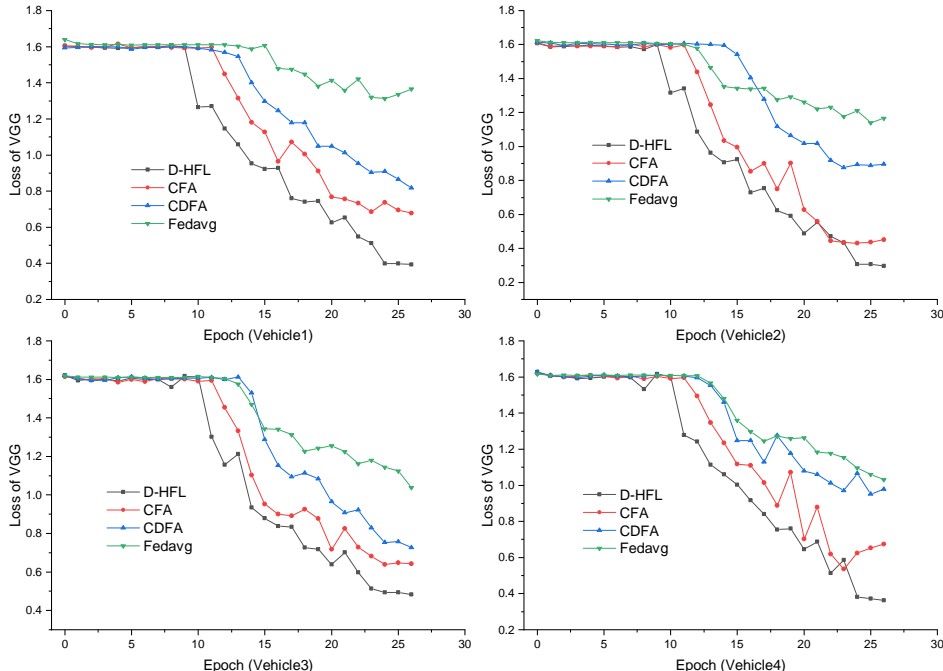

**Figure 6.** Loss of VGG.

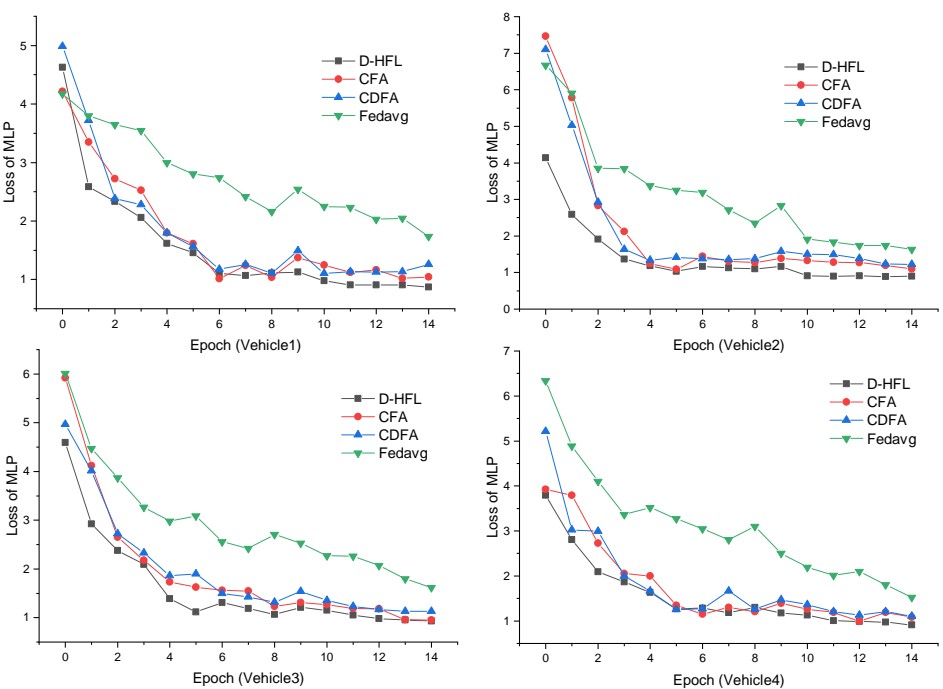

**Figure 7.** Loss of MLP.

We further analyze the accuracy of the models. As shown in Figure 8, all methods achieve an accuracy of over 65%, indicating that each method is capable of making correct predictions to a certain extent. This demonstrates the feasibility of modern machine learning approaches in advancing autonomous driving. Figure 8a displays the training accuracy of the VGG model under different distributed learning algorithms. As shown in the graph, the H-DFL algorithm achieves approximately a 9% improvement in accuracy compared to the CFA algorithm. Moreover, the improvement is more significant when compared to the FedAvg and CDFA algorithms. Similarly, Figure 8b illustrates the training accuracy of the MLP model under different distributed learning algorithms. The H-DFL algorithm also achieves higher accuracy performance.

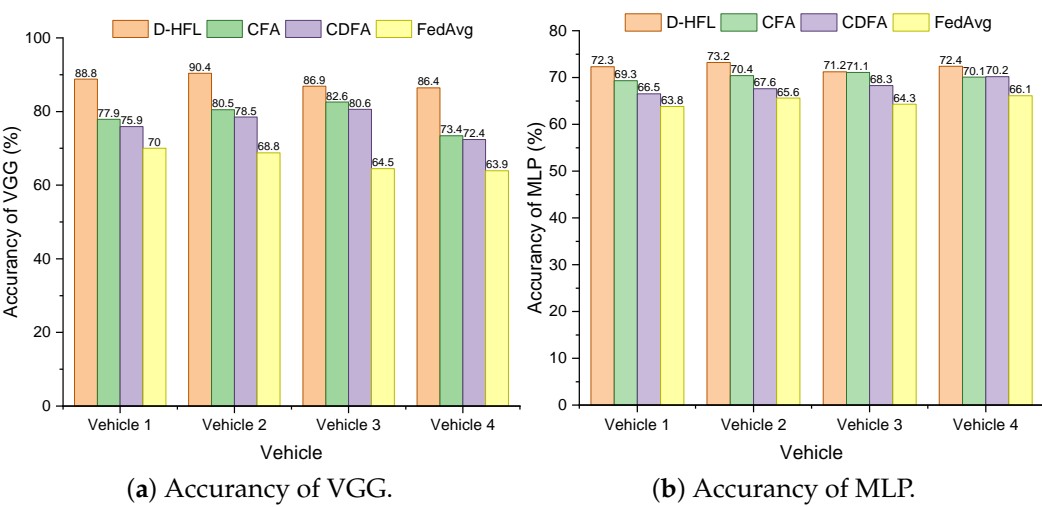

(**a**) Accurancy of VGG.　　　　　　(**b**) Accurancy of MLP.

**Figure 8.** Test accuracy (%) comparison of H-CFL to CFA, CDFA, and FedAvg.

In addressing non-IID data challenges, our framework's grouping strategy, based on Bloom filter, demonstrated remarkable capabilities in achieving data balance within clusters. The results showcased a significant improvement in the convergence of local models.

This outcome underscores the potential of Bloom-filter-based clustering to enhance the utilization of diverse data, leading to more robust and unbiased local models. Analyzing the trade-offs between accuracy and convergence speed revealed an interesting interplay. While the hierarchical decentralized federated learning approach achieved faster convergence due to the efficient cluster-level aggregation, there were subtle accuracy trade-offs compared to traditional methods that relied on central parameter updates. In some scenarios, the initial convergence speed of our framework was marginally slower, particularly when clusters contained fewer vehicles or when data distribution was extremely skewed. However, as training iterations progressed, our framework demonstrated an ability to catch up with and, in certain cases, surpass the accuracy of traditional methods. This suggests that, while there might be minor trade-offs in initial accuracy, our approach achieves competitive accuracy levels while benefiting from faster convergence in the long run.

The reason for these results is that the algorithm we proposed utilizes clustering based on data distribution, which mitigates the issue of data imbalance within clusters. On the other hand, CFA and CDFA algorithms only collaborate with neighboring vehicles, limiting their performance due to lower imbalance. The FedAvg algorithm involves training with all participants, resulting in a significant impact from data distribution imbalance, thereby reducing the efficiency of model learning.

## 6. Conclusions

By capitalizing on participant grouping and decentralized learning paradigms, H-DFL navigates the intricacies of data imbalances, accelerating convergence and enhancing model accuracy across various distributed learning algorithms. Moreover, the adaptable nature of H-DFL lends itself to diverse IoV scenarios. Its potential applicability in varied contexts, from urban to rural environments, differing vehicle densities, and contrasting data characteristics, underscores its versatility. As a clustering adaptive algorithm, H-DFL exhibits scalability and adaptability, poised to cater to the unique demands of different environments. By effectively addressing the challenges of model training and data sharing in vehicular networks, our framework has far-reaching implications for IoV applications, such as intelligent traffic management, collaborative safety measures, and dynamic decision-making. While the journey from theory to practice comes with its set of challenges, we are committed to addressing the potential barriers and aligning our framework with the practical nuances of real-world infrastructure.

**Author Contributions:** Conceptualization, S.L. and Z.L.; methodology, Z.X.; validation, W.L.; resources, J.T.; writing—original draft preparation, S.L.; writing—review and editing, Z.X.; supervision, Z.L. All authors have read and agreed to the published version of the manuscript.

**Funding:** This research was funded by the National Natural Science Foundation of China (61962044, 61962045), the Program for Young Talents of Science and Technology in Universities of Inner Mongolia Autonomous Region (NJYT23104), the Inner Mongolia Science and Technology Plan Project (2021GG0250), the Natural Science Foundation of Inner Mongolia Autonomous Region (2021MS06029), the Basic scientific research business fund project of universities directly under the autonomous region (JY20220324), and the Inner Mongolia Autonomous Region Higher Education Scientific Research Project (NJZZ22428).

**Data Availability Statement:** The data are available in a publicly accessible repository. The data presented in this study are openly available in BelgiumTSC at https://doi.org/10.1007/s00138-011-0391-3 (accessed on 2011), reference number [29] and CIFAR-10 at http://www.cs.toronto.edu/kriz/cifar.html (accessed on 2009), reference number [30].

**Conflicts of Interest:** The authors declare no conflict of interest.

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
