# Peer review of "Hierarchical Decentralized Federated Learning Framework with Adaptive Clustering: Bloom-Filter-Based Companions Choice for Learning Non-IID Data in IoV"

_electronics, doi:10.3390/electronics12183811_

Round 1

Reviewer 1 Report

see attached pdf

Minor editing of English language required

Reviewer 2 Report

Summary/Contribution: The novel framework Hierarchical Decentralized Federated Learning (H-DFL) for distributed model training and data sharing in vehicular networks is the main contribution of this work. H-DFL uses Bloom filters to compactly depict data complementarity for vehicle clustering to solve non-IID data issues. This allows cars to train their models on local data, exchange model parameters within each group, and reach certified local models without imbalanced data. H-DFL also uses a data center to obtain a global model with global features, and base stations update local models based on it. Real-world experiments show that H-DFL improves communication latency and solves vehicular network non-IID data issues.

Comments/Suggestions:

1. The introduction could be expanded to provide more context and motivation for the research. Consider adding a brief explanation of the importance and potential impact of vehicular networks and their challenges.

2. Clearly state the research problem or gap that this work aims to address. This will help set the expectations for the reader.
Methodology:

3. Provide a more detailed description of the methodology used in the proposed framework. Explain the steps involved in the hierarchical decentralized federated learning (H-DFL) process, including how vehicle clustering and data balancing are achieved.

4. Provide more information about the evaluation setup and metrics used to compare the proposed H-DFL framework against existing baselines.

5. Present the specific results and findings obtained from the evaluation in a clear and organized manner. Highlight the advantages or improvements of H-DFL over the baselines.

6. Expand the section on related work to provide a more comprehensive review of existing literature and approaches in the field of federated learning and vehicular networks.

7. Clearly summarize the strengths and limitations of previous approaches and how the proposed H-DFL framework addresses those limitations.

8. Emphasize the significance and potential impact of the H-DFL framework in addressing the challenges of model training and data sharing in vehicular networks.

9. Consider discussing future research directions or potential applications of the proposed framework.

10.  To increase the quality and significance of this paper, I suggest including a section on formal methods for AI-based technique verification. The authors might want to take into account the following sources that are pertinent to this subject:

a.https://ieeexplore.ieee.org/abstract/document/9842406

b. https://incose.onlinelibrary.wiley.com/doi/abs/10.1002/inst.12434

Can be improved

Reviewer 3 Report

The paper titled "Hierarchical Decentralized Federated Learning Framework with Adaptive Clustering: Bloom Filter-based Companions Choice for Learning Non-IID Data in IoV" introduces a comprehensive approach to address the challenges of decentralized model training in the context of Internet of Vehicles (IoV). The proposed Hierarchical Decentralized Federated Learning (H-DFL) framework with adaptive clustering and Bloom filter-based companions choice aims to improve data privacy and mitigate the effects of non-IID data distribution among vehicles. The topic is both relevant and promising; however, several aspects need further elaboration and clarification to enhance the paper's impact and comprehensibility.

  1. Introduction and Motivation: The introduction provides an overview of the challenges posed by IoV and motivates the need for decentralized approaches like Federated Learning. While the motivation is clear, it would be beneficial to elaborate on specific instances where traditional centralized methods fall short in addressing data privacy concerns. Additionally, the introduction could elaborate on the potential consequences of non-IID data distribution in vehicular networks, highlighting the significance of the proposed framework.

  2.  
  3. Hierarchical Decentralized Federated Learning (H-DFL): The description of the H-DFL framework is comprehensive, but some aspects require further clarification. For example, the relationship between vehicles, base stations, and data center servers should be visually represented or explained in more detail. Additionally, the process of how local models are updated based on the global model and the communication between these components should be expounded upon.

  4.  
  5. Bloom Filter-based Companions Choice: While the concept of using Bloom filter-based companions choice for data complementarity is intriguing, its details need more elucidation. How are these filters constructed? How is the choice of companions determined based on these filters? A step-by-step explanation or an example would help readers better grasp this critical aspect of the proposed framework.

  6.  
  7. Evaluation and Experimentation: The paper mentions "experimental results based on real-world data" to validate the proposed H-DFL framework. However, the evaluation methodology, datasets used, and evaluation metrics should be explicitly detailed. It's crucial to provide insights into how the experiments were conducted, including the setup, baseline methods for comparison, and statistical significance analysis.

  8.  
  9. Results and Discussion: The paper could benefit from a more in-depth analysis of the experimental results. Interpretation of the results should focus on addressing specific research questions, such as how the proposed framework mitigates non-IID data challenges, the extent of communication latency reduction, and the trade-offs involved in terms of accuracy and convergence speed.

  10.  
  11. Contributions and Future Work: The contributions of the paper should be summarized in the conclusion, highlighting the advancements made in addressing non-IID data challenges in IoV. Additionally, suggestions for potential extensions or future work could enhance the practical implications of the proposed framework.

  12.  
  13. Clarity and Language: Certain sentences and descriptions are complex and may benefit from simplification to improve readability. Clear and concise language will help readers understand the paper's concepts more easily.

  14.  
  15. References: Ensure that the references cited in the paper are accurately listed in the References section and that they cover relevant works in federated learning, decentralized approaches, and Internet of Vehicles. For example: you could benefit from

  16. 1) https://doi.org/10.3390/app13137745 2) https://ieeexplore.ieee.org/document/9625474  3) https://ieeexplore.ieee.org/abstract/document/9060868  4) https://ieeexplore.ieee.org/abstract/document/9479786?casa_token=RxPGGoMp6-AAAAAA:WRHhgtQfAXQ382Ax3qMfzV98FYHEI5n8mD2W-0KaIoJMHW4fhlaRJE8T2JdB6dftdFEMSTkscA

Addressing these comments will contribute to enhancing the paper's clarity, technical depth, and overall impact within the field of decentralized model training in vehicular networks.

Please see comments above. 

Round 2

Reviewer 1 Report

I want to thank authors for their fine work in answering my proposed revisions. I now believe that the article is greatly improved and I suggest its publication.

Reviewer 2 Report

The authors considered my comments and suggestions 

Can be improved 
